# Effect of Supplementation with Organic Selenium or Turmeric and Rosemary Mixture on Beta-Defensin Content in Goat Milk

**DOI:** 10.3390/ani12212948

**Published:** 2022-10-26

**Authors:** Magdalena Zalewska, Aleksandra Kapusta, Ewelina Kawecka-Grochocka, Daria M. Urbańska, Michał Czopowicz, Jarosław Kaba, Paulina Brzozowska, Emilia Bagnicka

**Affiliations:** 1Department of Bacterial Physiology, Institute of Microbiology, Faculty of Biology, University of Warsaw, Miecznikowa 1, 02-096 Warsaw, Poland; 2Department of Biotechnology and Nutrigenomics, Institute of Genetics and Animal Biotechnology PAS, Postępu 36A, 05-552 Jastrzębiec, Poland; 3Division of Veterinary Epidemiology and Economics, Institute of Veterinary Medicine, Warsaw University of Life Sciences-SGGW, Nowoursynowska 159c, 02-776 Warsaw, Poland

**Keywords:** beta-defensin, goat, herb, milk, organic selenium, spice, supplementation

## Abstract

**Simple Summary:**

At present, the growing interest of consumers in food products from animals raised in organic conditions has become evident. Combining organic farming with the highest care for animals’ welfare remains, however, extremely challenging. Functional food additives, such as herbs and spices, may enhance the immune system of animals and render them more resistant to pathogens. Furthermore, they may also increase animal productivity and positively modify milk properties. Goat milk is considered a healthy food for humans. Raising goats in accordance with strict organic farming principles may even further improve their quality. Therefore, it is crucial to develop and implement animal feeding systems that positively affect their productivity and reduce the need for the use of antibiotics. We hypothesize that adding herbs and spices or selenized yeast may enhance the resistance of the mammary gland to infections with pathogenic microorganisms by increasing the level of defensins in milk. Moreover, we investigate the effect of such functional food additives on the composition of goat milk.

**Abstract:**

The present study examines the effects of diet supplementation with an organic selenium or herb-spice mixture on beta-defensin1 (GBD-1) and beta-defensin2 (GBD-2) concentrations in goat milk. Herd-I, consisting of Polish White (PWI) and Fawn Improved (PFI) goats, received supplementation with organic or inorganic selenium (controls). All goats were free from parasites, mastitis, and small ruminant lentivirus infection. Herd-II, consisting of PWI goats, either received a turmeric-rosemary mixture (experimental) or not (controls). The Herd I control group demonstrated higher fat, free fatty acid, and somatic cell levels and lower protein and lactose contents than Herd II controls. The GBD-1 concentration was below the detection limit in both herds. Herd I controls demonstrated higher GBD-2 concentrations in milk than Herd II controls. In addition, lower GBD-2 concentrations were noted in PWI goat milk. Organic selenium or rosemary-turmeric mixture supplementation had no effect on the GBD-2 content in the milk of healthy goats. The higher GBD-2 concentration observed in Herd Ic than in Herd IIc may suggest that the type of basal diet affects defensin secretion.

## 1. Introduction

Defensins are important components of innate immunity, where they serve as the first line of defense against invading microorganisms. They have been identified in a wide range of organisms, including mammals, chickens, and turkeys. Defensins can be classified into three subfamilies, viz. α, β, and θ-defensins; however, the latter, a circular molecule also known as retrocyclin, is found only in old world monkeys, such as rhesus macaques, and orangutans [1,2]. Homologous transcripts of θ-defensins have also been identified in humans; however, they are not capable of forming amino acid chains due to the presence of a premature stop codon in the signal peptide [1].

Alpha- and β-defensins are small (2–5 kDa; 30–45 amino acids) cysteine-rich cationic peptides with a β-sheet structure stabilized by disulfide bonds. While all are known to exhibit activity against bacteria, fungi, and enveloped viruses [3,4,5], they differ with regard to the location of their intra-molecular disulfide bridges, the structure of their precursors, and their sites of expression [3]. Defensins are prevalent in cells and tissues directly involved in host defense against infections, such as granulocytic leukocytes, skin, mucosal surfaces, and other epithelia. In many animals, the highest concentrations of these peptides are found in granules: the storage organelles of leukocytes.

The α-defensins are involved in systemic and small intestinal host defense, and the β-defensins protect the mucosal epithelia of various organs [6]. The latter have been identified as within the lineage of innate and acquired immune responses. β-defensins are responsible for the recruitment of memory T cells and immature dendritic cells, and they also act as monocyte chemotaxis. Moreover, many studies have reported a link between the rapid expression of antimicrobial defensins and the recruitment of adaptive immune cells capable of directing a long-lasting cellular or humoral response to a pathogen. Additionally, β-defensins are very effective in promoting antigen-specific immune responses [7].

Defensins exert antimicrobial activity and cytotoxicity through the direct permeabilization of bacterial membranes and pore formation, resulting in cell content leakage and cell destruction [8]. In bacteria, membrane disruption coincides with the inhibition of RNA, DNA, and protein synthesis, and decreased bacterial viability. However, defensins may additionally inhibit bacterial cell wall synthesis and reduce the propagation of bacterial infections by neutralizing secreted toxins [9]. The less cationic defensins, such as human beta defensin 1 (HBD1) or human beta defensin 2 (HBD2), are more active against Gram-negative bacteria, while the more cationic forms, such as human beta defensin 3 (HBD3), efficiently neutralize fungi and both Gram-positive and -negative bacteria. Highly cationic peptides act against microorganisms in a structure-independent manner [10,11].

So far, most farm animals, including cattle and goats, have only been found to demonstrate β-defensins; however, they have not been observed in equines [12,13]. One study, based on a genomic screening strategy, identified 50 beta-defensin genes in the goat genome; these were densely clustered in four chromosomes (chromosomes 8, 13, 23, 27) with 16 detected by RT-qPCR [14]. However, goat beta-defensin 1 and 2 (*GBD-1* and *GBD-2*) genes have been studied in more depth. The *GBD-1* gene has been detected in the trachea, lungs, bronchi, and tongue, while *GBD-2* gene transcripts have been identified in the stomach, jejunum, small intestine, large intestine, and rectum [15,16,17]. Both peptides demonstrate activity against Gram-positive and Gram-negative bacteria [3].

During mastitis, beta-defensins are expressed to protect against bacterial invasion [16,18]. The infected tissues also demonstrated reduced concentrations of saline, which resulted in increased defensin activity. As salt is secreted into the milk during mammary gland inflammation, the milk develops a salty taste [19,20].

Many studies have found selenium plays a role in the immune response. It has a beneficial effect on animal health, improving the antioxidant effects of enzymes, reducing the possibility of mastitis or carcinogenesis, and promoting fertility. Selenium deficiency can impair phagocytosis and cause a range of diseases in farm animals [21], resulting in reduced milk yield, the development of mastitis, and, hence, elevated milk somatic cell count (SCC) [22,23,24]. Animals can receive selenium supplementation in different forms: inorganic (sodium selenite) and organic supplementation (selenized yeast), and via nanoparticles [21]. Although many studies have found organic selenium supplementation to have a more beneficial effect on milk yield than the inorganic form in both cows and goats [25], sodium selenite is the most commonly used supplement in goats and is included in many typical mineral mixtures [26].

Herbs and spices are also commonly known for their health-promoting effects. Moreover, they are affordable, readily accessible, and safe for health. Supplementation with turmeric has been associated with a reduced frequency of mastitis in dairy cattle, which has been attributed to the antioxidant and anti-inflammatory properties of elements found in curcuma (saponin, flavonoid, alkaloid, curcumin) [27]. Moreover, herbs containing saponins are believed to positively influence the microbiota in the rumen, optimizing the pH level in rumen fluid and subsequently lowering the number of potentially pathogenic bacteria. Other studies have found that turmeric paste accelerates wound healing and eliminates the probability of complications after skin injuries [28]. Dietary supplementation with 10% and 20% rosemary extract has been found to cause an increase in polyunsaturated fatty acid content in goat’s milk [29] and supplementation with cumin and curcuma to increase milk yield in cows [27].

Some researchers discovered GBD-1 in goat milk and demonstrated that the increased temperature of udder tissue during heat stress has no effect on the defensin concentration in milk [30], suggesting that it may continue to function even when the animal’s body temperature is elevated. However, our earlier studies yielded contradictory results regarding *GBD-1* and *GBD-2* expression at the mRNA level in milk. On the one hand, Jarczak [31] found increased expression of *GBD-2* in the milk cells of goats supplemented with active yeast *Saccharomyces cerevisiae*; however, no differences in expression were observed regarding the state of health of the mammary gland, i.e., between glands demonstrating a lack or presence of pathogen bacteria in milk. However, a study of organic selenium supplementation based on selenized yeast (*Saccharomyces cerevisiae*) on selected gene expression found *GBD-2* transcripts to be below the detection level in milk [25]. In both studies, the *GBD-1* transcripts in milk were below the detection level; however, it is known that the expression of the gene is typically not correlated with the presence of its protein product in milk. Therefore, the present study examines the concentrations of both peptides in goat milk. More specifically, the aim of the study was to determine the effect of a turmeric-rosemary mixture or selenized yeast supplementation on milk yield, milk technological parameters, and GBD-1 and GBD-2 concentrations in the milk of goats maintained in two herds. To achieve this, two separate experiments were performed.

## 2. Materials and Methods

### 2.1. Animals

Two experiments were carried out in two dairy goat herds in central Poland. All goats were healthy—herds were under constant veterinary control. Goats were free from small ruminant lentivirus (SRLV), as confirmed by multiple serological testing using ELISA (ID Screening ELISA, IDvet, Grabel, France) carried out every six months, free from parasitic infections using standard parasitological methods (floatation technique and McMaster method) and any signs of clinical or sub-clinical mastitis, as confirmed by veterinary examination, by a diplomate of the European College of Small Ruminant Health Management, and by acceptable upper limit of SCC in milk (<1.6 × 10^6^/mL), measured using Bactocount (IBCm, Bentley, Chaska, MN, USA). The goats were dewormed as needed. The experiments started in the third week of lactation, i.e., at the end of February, and lasted 80 days until the grazing period began. Morning milk samples were collected at the beginning and end of the study period.

The first herd (Herd I) consisted of about 50 dairy goats of two breeds: Polish White Improved (PWI) and Polish Fawn Improved (PFI). All animals were between their second and fifth lactation, and were kept in group pens (12 goats in each). The mean body weight was approximately 50 kg, and the mean daily milk yield was 3 kg. Goats were fed once a day with maize silage, hay, and oat straw ad libitum. They also received approximately 1 kg concentrates, in accordance with the system developed by the French National Institute for Agricultural Research (Institut National de la Recherche Agronomique, INRA), France, and adopted by the National Research Institute of Animal Production, Poland [32]. Water and salt lick were available at will. From the herd, 24 animals were selected and divided into two groups: experimental (Herd Ie), supplemented with organic selenium (N = 12) and control (Herd Ic), supplemented with inorganic selenium (N = 12). The two groups were identical in terms of breed (N = 6 of PWI and N = 6 of PFI in each group) and parity (between second and fourth lactation). Goats assigned to the experimental group were fed the mineral-vitamin mixture without inorganic selenium additives, specially prepared for this experiment by Polmass (Bydgoszcz, Poland). The goats also received starch capsules containing selenized yeast (approximately 0.6 mg of selenium/day/goat) (Se-yeast, Sel-Plex 1000, Alltech, Warsaw, Poland) fed in the evening milking into individual troughs. Consumption of the capsules was monitored by the caretaker. The control group was routinely supplemented with inorganic selenium included in a mineral-vitamin mixture: 45 mg of sodium selenite/kg of the mixture (approximately 0.7 mg of selenium/goat/day) (Vitamix C, Polmass, Bydgoszcz, Poland), mixed with concentrated feed. The details of basal diet and selenized yeast have been given previously [25].

The second herd (Herd II) included approximately 100 goats from the PWI breed. Of these, 30 goats were included in the study; these were divided into control (N = 15) (Herd IIc) and experimental (N = 15) (Herd IIe) groups. The groups were identical with respect to parity (i.e., between third and fifth). All animals were fed with haylage (approx. 3 kg), concentrates (approx. 1 kg), and barley straw. The diet of the experimental group was further supplemented with turmeric and rosemary mixture (Selko^®^ AOmix, Trouw Nutrition Polska sp. z o.o., Grodzisk Mazowiecki, Poland), given orally in capsules during milking time (approximately 0.7 g/goat/day) by the caretaker.

### 2.2. Methods

Milk samples were collected in the morning prior to milking. Teat surfaces were washed with disinfectant (70% ethanol); then, before the milk sampling, a small amount of foremilk was discarded into a pre-milk cup. The next two sets of representative milk samples (30 mL from the whole udder milk mixed before collection) were taken. One set of tubes contained preservatives, i.e., Microtabs (Bentley, Chaska, MN, USA), to study milk composition. The other samples, without any additives, were used for the defensin concentration investigation. Milk samples were stored at 4 °C prior to tests and as soon as possible after collection (max next day) and were subjected to analyses described below.

The milk without additives was skimmed by centrifugation for 20 min at 2000 rpm at 20 °C. It was then tested for GBD-1 and GBD-2 levels using ELISA tests (Wuhan Fine Biotech Co., Wuhan, Hubei, China) according to the manufacturer’s protocol.

Milk composition (fat, protein, casein, lactose, total solids (TS), solids non-fat (SNF), urea, free fatty acids (FFA), freezing point depression (FPD), citric acid contents, total acidity, and density) was measured using a MilkoScan FT analyzer (FOSS Analytical A/S, Hillerød, Denmark). Milk SCC levels were determined with the Bactocount IMCm (Bentley, Chaska, MN, USA).

### 2.3. Statistical Analysis

The normality of data distribution was checked using the UNIVARIATE (SAS/STAT packages 2002–2012, version 9.4) procedure. Before the statistical analysis, the distribution of the data was normalized by transforming the SCC values into a natural logarithm scale (lnSCC). Three multi-factorial ANOVA tests (analysis of variance) were conducted separately using SAS/STAT packages (2002–2012, version 9.4; SAS Institute, Raleigh, NC, USA). In the first analysis, the differences in milk yield and its components, including defensin contents, were compared between control groups, with breed and parity being included in the model as fixed effects; however, the herds did not share identical breed profiles. The second analysis included only the data obtained from Herd I, identifying differences in defensin levels between the experimental (organic selenium) and control (inorganic selenium) groups and between breeds. The third analysis evaluated the differences between the experimental (turmeric-rosemary mixture) and control (no supplementation) groups in Herd II. As preliminary research suggested that the point of sampling was not significant in the second experiment (i.e., turmeric-rosemary mixture), this effect was not included in the statistical analysis.

The Pearson’s correlation coefficient was estimated between the concentrations of defensins and other milk components, and the milk yield among the control groups from the two herds. This was also performed using the PROC CORR of the SAS/STAT package. The significance level was set at *p* ≤ 0.05 (*) and *p* ≤ 0.01 (**).

## 3. Results and Discussion

A number of differences in milk yield and composition were found between the studied control groups; however, no changes were observed for casein, SNP, citric acid content, or acidity (Table 1). The Herd I control group had higher fat, FFA, and lnSCC levels (*p* ≤ 0.05) but lower lactose contents (*p* ≤ 0.05) than the Herd II controls; however, no differences in milk traits were observed between the PWI and PFI breeds in Herd I. This suggests that the goats in Herd II had healthier mammary glands than Herd I. A lower FPD was stated in Herd I. This parameter depends on many factors, with the main ones being milk components (lactose, chlorides, calcium, potassium, magnesium, phosphates), feed and water intake, and the health of the mammary gland [33]. FPD appears to be lower in goat milk than in cow milk, which can vary between −0.520 and −0.600 °C [34]; however, the co-occurrence of FPD values below −0.600 °C with low lactose content can indicate mastitic problems [35].

A higher level of FFA was observed in Herd I, which may indicate an increased level of free radical lipid peroxidation in milk [36,37]. Large differences in fat content were noted, which indicates significant differences in diet compositions; dietary fiber intake is essential for regulating fat content in milk, as this depends on the acetate to propionate ratio in the rumen, with acetate being the major precursor of milk fat. Nutritionally adequate content is 18–20% acid detergent fiber (ADF) or 41% neutral detergent fiber (NDF) in the diet. An increase in ADF intake from 396 to 839 g/day was found to cause an increase from 4.85 to 5.40% of fat in goat milk [38,39]. The fat:protein ratio also indicated that the diet consumed by the goats in Herd II was not the same as theoretically balanced and prepared; this ratio is commonly used as an indicator of energy deficit, subclinical ketosis, or subacute ruminal acidosis (SARA) in cattle. Excessive urea content was noted in the milk of goats from Herd II. This also indicates improper energy and protein balance in the diet [40].

The goats in Herd I have been inseminated with French Saanen and Alpine buck semen (depending on the does breed) for many generations; however, only PWI bucks have been used in Herd II, including some of those from Herd I. Although no differences in milk, fat, or protein yields were previously noted between French Saanen, French Alpine, PWI, and PFI breeds in the entire Polish active population [41], there may still be some genetic distance between those two herds. It should be stressed that all goats used in the studies had an acceptable upper level of SCC, without any symptoms of mastitis, and were free of SRLV.

In both herds, parity appeared to have no impact on most studied milk traits. However, many previous studies have found milk yield to increase and milk components to decrease until the fourth lactation [41,42,43]. In addition, LnSCC was found to be higher during later lactations in both herds (Herd I: 4.47. 6.24, 6.20, SE = 0.31; Herd II: 5.04, 6.01, 6.12, SE = 0.35), which is also consistent with previous studies [44,45]. In Herd II, FFA also increased with lactation number (0.48, 0.61, 0.74, SE = 0.05). Despite this, parity had no effect on the GBD-2 peptide content.

The effect of selenium yeast supplementation on the productivity traits in Herd I at the beginning and end of the experiment is presented in Table 2. Differences in milk yield, lnSCC, and FPD were observed between the groups at the beginning and the end of the experiment. According to Zhang et al. [46], dietary Se supplementation increases milk production, with organic Se being more effective than inorganic Se. Additionally, selenium supplementation raised the levels of selenium and glutathione peroxidase in whole blood, with organic selenium being more efficient than inorganic selenium. Similarly, Reczyńska et al. [25] found milk, fat, and protein yields to be approximately 5%, 24%, and 17% higher in goats treated with organic Se; however, this was performed throughout the entire lactation period.

Maternal supplementation with Se and iodine (I) as a slow-release ruminal bolus applied in late pregnancy improved both the milk production of grazing goats and the performance of their offspring [47]. Se concentrations in colostrum were higher in goats supplemented with L-selenomethionine during pregnancy than in animals supplemented with sodium selenite or selenium nanoparticles. Kids of does receiving L-selenomethionine had higher whole-blood and serum Se concentrations when compared to kids of goats from other groups. Nano-selenium was unable to raise newborn Se concentrations above control levels [48].

In Herd II, no significant differences in milk yield or composition were recorded between the control and experimental groups (Table 3). Similar results have been demonstrated by Boutoial et al. [29], who found that the addition of rosemary to the basal diet did not affect the chemical composition of goat milk. However, Chiofalo et al. [49] report that sheep supplemented with different doses of rosemary (600 and 1200 mg/day/ewe) had higher milk yield than untreated controls, with the highest performance demonstrated by sheep receiving the highest dose of rosemary; the same relationship was observed for fat, protein, casein, and lactose content.

Many food additives have been tested for their beneficial properties. Hashemzadeh-Cigari et al. [50] found supplementation with a mixture of 60% rosemary, 18% cinnamon bark, 18% turmeric, and 4% clove bud to have a beneficial impact on the health state of the mammary gland in dairy cows, but only in those with high SCC. In contrast, in the present study, all animals involved were free from any signs of mastitis, including subclinical infection, as the average SCC reached an acceptable upper limit in all groups from both herds. Abdullah et al. [51] revealed that lemongrass and roselle herbs supplementation improved milk composition and rumen fermentation without lowering animal performance. Compared to basil essential oil, marjoram essential oil supplementation improved the digestibility of nutrients (dry matter and organic matter), ruminal fermentation (propionic, valeric, and total short-chain fatty acids concentration), milk production, and composition (fat percent, fat corrected milk (4% fat; FCM) and energy corrected milk yields, total solids, fat, and lactose yields, milk efficiency), according to Tawab et al. [52]. Similar results were obtained by Kholif et al. [53] with supplementation of the lactating Boer goats diet with Chlorella vulgaris microalgae with or without copper, as well as Ghoneem and Mahmoud [54] who found that the supplementation of lactating goat’s diet with thyme leaves or essential oil improved the digestibility but without influencing rumen fermentation parameters. Moreover, the addition of thyme leaves increased milk and FCM yields and total unsaturated fatty acid concentrations while decreasing total saturated fatty acid levels. Simultaneously, supplementation lowered cholesterol and triglyceride blood concentrations.

It should be stressed that the supplementation’s effect may depend on the goat breed. Amosu et al. [55] and Oderinwale [56] proved this regarding overall growth performance during the gestation period and oxidative stress and cortisol concentration, respectively, after adding turmeric powder to the diet. Moreover, it should be stressed that the goats used in the study were clinically healthy. Therefore, it is possible that a rosemary-tumeric mixture full of natural polyphenols, which are strong antioxidants, does not affect the goats’ organism if their homeostasis is not disturbed, but it might influence goats with disease problems.

While the GBD-1 level was below the detection limit in all goat milk samples from both herds, GBD-2 concentration differed (*p* < 0.01) between the control groups from the two herds (Figure 1). No comparison of GBD-2 concentrations was made between the two experimental groups, as two different factors were used.

A higher GBD-2 concentration was noted in Herd Ic than in Herd IIc. This may mean that the homeostasis of the udder of the goats in Herd I may have been disturbed. Alternatively, the differences may have been caused by the disparity between the basal diets: Herd I received corn silage, while Herd II received dry pellets and grass silage. However, both herds were fed in accordance with the INRA system adopted by the Research Institute of Animal Production (IZ PIB), Poland [32].

It is also possible that goats’ homeostasis could also have been disturbed by changes in food habits caused by social hierarchy behavior. Goats tend to step aside and stop feeding when a neighbor is socially dominant. The dominant animals therefore have greater access to food, both when feeding free in the pasture, and when taking the food in the stall. This would affect the average group score. While this effect is less prominent in pastures, it can have a lot of influence in a limited space. The animals of intermediate rank in the group are the most productive, since neither are they pressed as subordinates nor do they have to intervene constantly to maintain their status, like the dominant ones [57]. Unlike Herd I, the goats in Herd II were maintained in a large barn with less hierarchical pressure. However, this is just speculation, and further research on the hierarchical structure within these herds is necessary.

No *GBD-2* gene expression was noted in a previous study of the same herd [25]. It is possible that the transcript level simply did not reach the level of detection by RT-qPCR in the control or experimental groups; indeed, Jarczak [31] did not detect *GBD-1* gene transcripts in milk somatic cells from the same breeds as the present study, but did detect *GBD-2* mRNA. This is in line with both of our present results. Moreover, they indicate that supplementation with organic and inorganic selenium influenced on β-defensin gene expression in dairy cows [58]. It is also possible that the environmental conditions (e.g., harsh weather during winter in a moderate climate zone: transitional character between maritime and continental types) may also have had an effect on the defensin genes; however, some evidence suggests a low correlation between transcript level and the concentration of the protein product [59]. These previous studies focused only on the mRNA level and not on defensin concentration. It is also possible that the selenium status of the research animals affected our results. All goats in Herd I have been routinely supplemented with mineral-vitamin mixture with inorganic selenium, and they probably did not suffer the deficiency of this microelement. If they have a selenium deficiency, its impact on defensin concentration could be greater. However, the formation of a group with a Se deficit is impossible under production conditions.

The absence of the GBD-1 peptide in our study suggests that it does not play a role in maintaining udder health, and that the added supplements did not affect its content in the milk. A high level of GBD-2 was observed in the milk of all studied groups (1686–2886 pg/mL); however, no differences in GBD-2 concentrations were noted after supplementation with either selenium or the herb-spice mixture. It seems that this defensin is present in the milk of healthy goats, but the supplements did not affect its concentration. However, it is possible that the tested supplements may have a noticeable impact on diseased animals. Hashemzadeh-Cigari et al. [50] found rosemary supplementation to have a beneficial impact on mammary gland health but only in cows with high SCC, indicating mammary gland inflammation. It has already been proven that toll-like receptors (TLRs) promote the release of defensins by triggering antimicrobial pathways. TLRs are not active in healthy goats [60,61], so they cannot stimulate the production of defensins. However, the presence of GBD-2 and the lack of GBD-1 may indicate that GBD-2 is the first line of mammary gland defense against pathogens and that the udder is prepared for possible intrusion.

Regarding the GBD-2 concentration in the two breeds in Herd I, concentration was found to be higher in PFI than PWI milk (Figure 2) (3385.77 ± SE = 479.72 and 2127.69 ± SE = 831.65 pg/mL, respectively; *p* ≤ 0.01). Sharma et al. [62] proposed that breed may also be a factor in determining defensin content. This was found in the case of the LAP gene, which is detectable only in the Indian goat genome [15]. However, in a study of two main Polish goat breeds, Plawińska-Czarnak et al. [63] did not find any differences in the expression of 44,000 genes, including those involved in the immune system and productivity traits, in the milk somatic cells, milk fat globules, or peripheral blood leukocytes. Notwithstanding, the question arises as to whether the defensins were included among the analyzed genes. Indeed, Chen et al. [64] found higher *pBD-1, 2,* and *3* expression in Meishan pigs than crossbred ones. Thus, further studies on the differences in defensin gene expression between dairy goat breeds are needed.

Both positive and negative correlations were observed between GBD-2 and most studied milk traits, with most being moderate or high (Table 4). Of the 14 measured parameters, only lnSCC, casein, SNF, citric acid, and acidity were not correlated with the GBD-2 concentration. This might indicate that there is no relationship between GBD-2 and the goat udder health state, but on the other hand, it may be involved in the maintenance of homeostasis. A positive correlation was observed between GBD-2 and milk yield, which suggests that a higher concentration of defensins may be expected in milk of highly productive goats.

However, while no correlation was observed between lnSCC and the GBD-2 level in the present study, Roosen et al. [18] and Ryniewicz et al. [65] report a positive correlation between goat β-defensin gene expression and SCC level. This is to be expected, as defensin genes are triggered in response to inflammation, and it is commonly known that udder inflammation results in an increase in milk SCC [66,67]. However, as stressed above, all animals involved in the study were free from any sign of mastitis, and SCC in goat milk is influenced by many effects, except the presence of pathogenic bacteria [68].

The GBD-2 level was also found to be positively correlated with fat and FFA content, and negatively correlated with protein content. This might indicate an association with oxidative stress, as higher levels of fat and FFA together are indicators of lipid peroxidation, resulting in protein reduction [36]. Moreover, protein content indicates the suitability of milk for processing. As defensin content is typically increased in inflammatory states, milk from diseased goats has a worse technological quality [66]. In addition to its negative correlation with protein content, GBD-2 also showed a high negative correlation with lactose content and a positive correlation with FPD. This indicates that a higher concentration of GBD-2 is associated with poorer udder health status: low lactose content and an FPD value closer to zero indicate poor health status [35].

## 4. Conclusions

Neither organic selenium nor rosemary-turmeric mixture supplementation appeared to have any effect on the GBD-2 content in the milk of healthy goats. Our findings suggest that its level in milk may be associated with the breed of goat; however, these results should be confirmed in further studies. Parity did not influence the GBD-2 concentration in milk.

The presence of GBD-2 in the milk of healthy goats, combined with a lack of GBD-1, indicates that GBD-2 might play an important role in the first line of defense against pathogens in milk, or maybe that these two defensins play different roles in the mammary gland. Again, further study is needed to understand the role of defensins in maintaining homeostasis in the udder.

## Figures and Tables

**Figure 1 animals-12-02948-f001:**
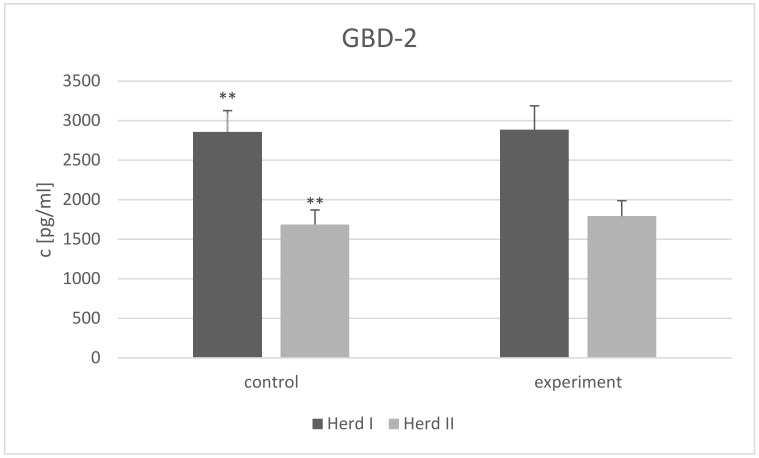
β2-defensin 2 (GBD-2) concentration (c) (presented as LSMEAN value with standard errors) in the milk of goats from two herds; Herd I control group—supplemented with inorganic selenium (sodium selenite), Herd II control group—without any supplementation, Herd I experimental group—supplemented with organic selenium (selenized yeast), Herd II experimental group—supplemented with spice-herb mixture, ** differences between the control groups from both herds *p* < 0.01.

**Figure 2 animals-12-02948-f002:**
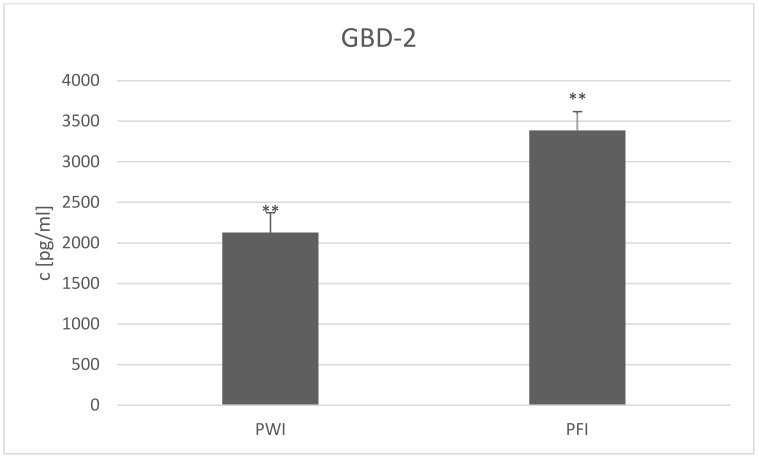
β-Defensin 2 (GBD-2) concentration (c) (presented as LSMEAN value with standard errors) in milk of PWI and PFI breeds from Herd Ie at the end of the experiment; PWI—Polish White Improved; PFI—Polish Fawn Improved; ** differences between breeds at *p* ≤ 0.01.

**Table 1 animals-12-02948-t001:** The differences in the yield and composition of milk from morning milking between the control groups of the two herds, regardless of the time of sampling.

Trait	Herd I		Herd II	
	LSM	SE	LSM	SE
Milk yield [kg/morning milking]	2.35 ^A^	0.17	1.46 ^B^	0.13
LnSCC^#^ ≈ SCC	6.98 ≈ 1.08 × 10^6 A^	0.41	5.04 ≈ 1.55 × 10^5 B^	0.31
Fat [%]	4.31 ^A^	0.23	2.17 ^B^	0.18
Protein [%]	2.79 ^A^	0.08	3.20 ^B^	0.06
Casein [%]	2.13	0.08	2.38	0.06
Lactose [%]	4.66 ^A^	0.06	4.99 ^B^	0.05
TS [%]	12.31 ^A^	0.27	10.64 ^B^	0.21
SNF [%]	8.25	0.12	8.47	0.09
Urea [mg/L]	157 ^A^	26	445 ^B^	20
Citric acid [%]	0.11	0.07	0.10	0.05
FPD [°C]	−636 ^A^	6.01	−596 ^B^	6.00
FFA [mEKV/L]	1.10 ^A^	0.06	0.57 ^B^	0.05
Density [mg/mL]	1024 ^A^	0.58	1027 ^B^	0.48
Acidity [T] *	16.68	0.59	16.25	0.59

LSM—least-square means; SE—standard errors; LnSCC^#^—natural logarithm of somatic cell count in milk; TS—total solid; SNF—solid non-fat; FPD—freezing point depression; FFA—free fatty acids; * Turner degrees of acidity; A,B—different letters within rows means differences at *p* ≤ 0.01.

**Table 2 animals-12-02948-t002:** Yield and composition of milk from morning milking in Herd I at the beginning and end of the experiment.

Trait	Sampling Time	Control Group	Experimental Group	SE
		LSM	LSM	
Milk yield [kg/morning milking]	beginning	1.92 ^A^	2.06 ^A^	0.10
	end	2.46 ^Ba^	2.80 ^Bb^	
LnSCC^#^ ≈ SCC	beginning	5.85 ≈ 3.50 × 10^5^	5.55 ≈ 2.58 × 10^5^	0.10
	end	6.17 a ≈ 4.80 × 10^5^	5.22 b ≈ 1.85 × 10^5^	
Fat [%]	beginning	5.21 ^A^	4.85 ^A^	0.14
	end	3.94	3.80 ^B^	
Protein [%]	beginning	3.09	3.01	0.06
	end	2.99	2.97	
Casein [%]	beginning	2.35	2.30	0.05
	end	2.29	2.24	
Lactose [%]	beginning	4.79 ^A^	4.77 ^A^	0.04
	end	4.63 ^B^	4.46 ^B^	
TS [%]	beginning	13.76 ^A^	13.29 ^A^	0.18
	end	12.12 ^B^	11.80 ^B^	
SNF [%]	beginning	8.70 ^A^	8.62 ^A^	0.08
	end	8.40 ^B^	8.1 ^B^	
Urea [mg/L]	beginning	91.60 ^A^	72.42 ^A^	12.9
	end	155.55 ^B^	146.21 ^B^	
Citric acid [%]	beginning	0.09	0.09	0.005
	end	0.10	0.09	
FPD [−°C]	beginning	−598 ^A^	−589 ^A^	4.83
	end	−630 ^Ba^	−609 ^Bb^	
FFA [mEKV/L]	beginning	1.35 ^A^	1.26 ^A^	0.07
	end	0.96 ^B^	0.85 ^B^	
Density [mg/mL]	beginning	1025.7	1025.4 ^A^	0.32
	end	1024.8	1024.1 ^B^	
Acidity [T] *	beginning	15.28	15.54	0.47
	end	16.28	16.05	

LSM—least-square means; SE—standard errors; LnSCC^#^—natural logarithm of somatic cell count in milk (the absolute value approx. between 130 and 160 × 10^3^); TS—total solid; SNF—solid non-fat; FPD –freezing point depression; FFA—free fatty acids; * Turner degrees of acidity; A,B—different letters within rows means differences at *p* ≤ 0.01; a, b—different letters within rows means differences at *p* ≤ 0.05.

**Table 3 animals-12-02948-t003:** The yield and composition of milk from morning milking in Herd II at the beginning and end of the experiment.

Trait	Sampling Time	Control Group	Experimental Group	SE
		LSM	LSM	
Milk yield [kg/morning milking]	beginning	1.24	1.26	0.13
	end	1.38	1.37	
LnSCC^#^ ≈ SCC	beginning	5.33 ≈ 2.07 × 10^5^	4.75 ≈ 1.16 × 10^5^	0.31
	end	5.44 ≈ 2.32 × 10^5^	5.17 ≈ 1.76 × 10^5^	
Fat [%]	beginning	1.58 ^A^	1.92 ^a^	0.17
	end	2.36 ^B^	2.57 ^b^	
Protein [%]	beginning	3.08	3.03	0.05
	end	3.15	3.06	
Casein [%]	beginning	2.24	2.32	0.05
	end	2.35	2.27	
Lactose [%]	beginning	5.34 ^A^	5.24 ^A^	0.05
	end	4.97 ^B^	4.95 ^B^	
TS [%]	beginning	10.23	10.45	0.17
	end	10.78	10.89	
SNF [%]	beginning	8.76 ^a^	8.64 ^a^	0.07
	end	8.42 ^b^	8.32 ^b^	
Urea [mg/L]	beginning	307 ^A^	327 ^A^	15.48
	end	455 ^B^	426 ^B^	
Citric acid [%]	beginning	0.14 ^A^	0.13 ^A^	0.005
	end	0.10 ^B^	0.09 ^B^	
FPD [°C]	beginning	−628 ^A^	−615 ^a^	3.32
	end	−599 ^B^	−601 ^b^	
FFA [mEKV/L]	beginning	0.50	0.55	0.04
	end	0.60	0.59	
Density [mg/mL]	beginning	1027.4	1026.9	0.40
	end	1026.8	1026.4	
Acidity [T] *	beginning	13.82 ^A^	13.88 ^A^	0.41
	end	16.29 ^B^	15.85 ^B^	

LSM—least-square means; SE—standard errors; LnSCC^#^—natural logarithm of somatic cell count in the milk (the absolute value approx. between 130 and 160 × 10^3^); TS—total solid; SNF—solid non-fat; FPD—freezing point depression; FFA—free fatty acids; * Turner degrees of acidity; A,B—different letters within rows means differences at *p* ≤ 0.01; a, b—different letters within rows means differences at *p* ≤ 0.05.

**Table 4 animals-12-02948-t004:** Correlations between beta-defensin concentrations in milk with components and yield.

Trait	GBD-2
Milk yield [kg/morning milking]	0.42 **
LnSCC ^#^	-
Fat [%]	0.56 **
Protein [%]	−0.33 *
Casein [%]	-
Lactose [%]	−0.66 **
TS [%]	0.43 **
SNF [%]	-
Urea [mg/L]	−0.48 **
Citric acid [%]	-
FPD [°C]	0.31 *
FFA [mEKV/L]	0.49 **
Density [mg/mL]	−0.47 **
Acidity [T] ^##^	-

* *p* ≤ 0.05; ** *p* ≤ 0.01; ^#^ LnSCC- natural logarithm of somatic cell count in milk; TS—total solid; SNF—solid non-fat; FPD –freezing point depression; FFA—free fatty acid content; ^##^ Turner degrees of acidity.

## Data Availability

The datasets used and/or analyzed during the current study are available from the corresponding author upon reasonable request.

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
