# Peer review of "Effect of Supplementation with Organic Selenium or Turmeric and Rosemary Mixture on Beta-Defensin Content in Goat Milk"

_animals, 2022, doi:10.3390/ani12212948_

Round 1
Reviewer 1 Report
Animals: "Effect of supplementation with organic selenium or turmeric and rosemary mixture on beta-defensin content in goats” by Zalewska et al.
The authors studied the effects of adding herb and spices vs. yeast with selenium on dairy goats on two farms. For Herd 1, 24 lactating does were selected and fed their regular diet, with or without added selenized yeast. In Herd 2, 30 lactating does were divided into 2 groups representing regular diet with or without added rosemary-turmeric. Goats were studied beginning the 3rd week of lactation for a period of 80 days. The main purpose of the study was to determine the effect of these 2 dietary additions on the content of beta-defensins (1- and 2-; GBD-1, GBD-2) in goat milk. Additionally, several milk components were measured as well, such as SCC, protein, fat, etc. GBD-1 was not detected in any of the milks. Neither organic selenium nor rosemary-turmeric supplementation was related to any change in GBD-2 concentration in milk. The authors concluded that several areas related to GDB in goat milk need further study.
In general, the manuscript reads fairly well and is organized appropriately. More information is needed to evaluate some aspects of the manuscript, as noted hereafter.
Specific comments:
ll. 13-24: There is a lot of “howevers” in this section. Perhaps you can modify or use another word.
l. 14: It is not entirely clear to me what “ecological farming” is? Is there another way that this could be stated?
ll. 21-24: Is it intentional that you do not mention selenium here?
ll. 83-85: “…have been identified found ….” Seems like just one of the 2 words can be used here (vs. both): Identified vs. found.
l. 118: “…on antibiotic use, more specifically, their complete abandonment.” This is kind of an awkward statement. Can’t you simply state something like this: “Organic farming involves strict restrictions on antibiotic use, to the point of prohibiting their use”? There are variations among countries in regulations in terms of antibiotic use in animals producing organic products. Organic dairy product production in the U.S. involves complete prohibition on antibiotic use. My understanding is that some countries in western Europe allow antibiotic use, but with an extended withdrawal time. I am not sure of the regulations in Poland. You need to make sure your statements consider these variations—or are stated with reference to the specific country denoted. Thus, the statement above could be modified to: “Organic farming IN POLAND….”
ll. 140-144: You state the goats are free SRLV and mastitis? What about any other diseases and parasites?
l. 144: Your inclusion criteria include “low SCC….” Could you quantify what was considered as a low SCC?
ll. 146-147: More details should be provided on when and how milk samples were collected. Were they collected prior to, or after, milking? What preparation was done to the teats? How were samples handled after collection (refrigerated, etc.)?
ll. 157 and ll. 168-170: Could you explain how you selected the goats for these 2 experiments?
ll. 161-167 and ll. 172-174: How confident are you that each goat got approximately the desired amount of selenium per day?
l. 170: “The groups identical with respect to..” I believe you mean to state: “The groups WERE identical with respect to…”
l. 247: As the start of a new sentence, “despite” should be “Despite.”
ll. 302-312: You may want to consider that the “selenium status” of the research animals could affect the results you obtain. For example, additional selenium may have little impact on animals with adequate selenium stores, but could have a much greater impact on animals with low selenium stores.
l. 329: “Tool-like receptors should be “toll-like receptors.”
ll. 351-373: You discuss correlation more than once here. You do interpret the meaning “conservatively” with statements such as “may suggest” or “might indicate.” Remember, as you know, that correlation does not assess “cause” and “effect.”
Figure 1: The way you designate that the 2 control groups differ is awkward. Not sure the best way to handle it. One option might be to reverse the bars for the experimental and control groups for herd 1, leaving the 2 control groups next to each other. You may have better idea(s) in terms of how to do this.
Author Response
Please, find our explanations in the attached file

Reviewer 2 Report
Dear Authors,
animals-1895477
Effect of supplementation with organic selenium or turmeric 2 and rosemary mixture on beta-defensin content in goat milk
The scientific issue of the manuscript is particularly relevant and responds to the topics of the journal. The relevance and the originality of the topic makes this manuscript worth to be published. The topic is well documented and include good tables and figures, However, there are some remarks and suggestions that should be taken into account by the authors, as follows.
Introduction is too large. I suggest to reduce it and try to focus it to the topic of the study.
It is not necessary to include so ample justification, line 115-122 could be deleted.
Table 1, 2 and 3. Add either, in the title or as a footnote, the meaning of LSM and SE.
Figure 1 and 2. Please add units on Y axis.
Figure 1 and 2. “Herd 1 c” and “Herd 1 b” could be changed to “Herd 1c” and “Herd 1e” etc. the same on the rest of the manuscript.
Please add recent references on the topic of the study (30% between 2017 and 2022).
Author Response

(The authors gave the same response as above.)

Round 2
Reviewer 1 Report
Animals, Revision 1: Effect of supplementation with organic selenium or turmeric and rosemary mixture on beta-defensin content in goats” by Zalewska et al.
This is a revision of the original version of this manuscript, after responding to reviewer comments. The authors studied the effects of adding herb and spices vs. yeast with selenium on dairy goats on two farms. For Herd 1, 24 lactating does were selected and fed their regular diet, with or without added selenized yeast. In Herd 2, 30 lactating does were divided into 2 groups representing regular diet with or without added rosemary-turmeric. Goats were studied for a period of 80 days. The main purpose of the study was to determine the effect of these 2 dietary additions on the content of beta-defensins (1- and 2-; GBD-1, GBD-2) and other milk components in goat milk. GBD-1 was not detected in any of the milks. Neither organic selenium nor rosemary-turmeric supplementation was related to any change in GBD-2 concentration in milk. The authors concluded that several areas need further study.
In general, the manuscript reads fairly well and is organized properly. The authors have responded appropriately to most comments from the initial review. Specific comments to consider include the following:
Specific comments:
ll. 139-142: Please indicate how the veterinarian determined there were no parasites. “…by veterinary examination conducted by a diplomate of the European College of Small Ruminant Health Management….” You characterize a SCC of <1.6 million/mL as a low SCC. In my opinion, this is not really a low SCC—I would characterize it as “acceptable upper limit.”
ll. 157-160: Modify verb to: “Goats assigned to the experimental group were (not was) fed….” You could consider ending one sentence after Polmass (Bydgoszcz, Poland). Then start another: “The goats also received starch capsules..”
l. 161: Here you say “served.” You could consider replacing that with “fed.”
ll. 172-174: An important point in any study involving products like turmeric and rosemary is to give as much detail as you can in terms of the source, identity, purity, and preparation methods of the products. As we all remember, the idea is to give the reader as much information as possible to be able to reproduce the study.
l. 194: Give the source of the UNIVARIATE program—I assume SAS/STAT, but you should specify.
ll. 234 and 238: You use “actually taken” and “taken” here. I think it would sound better to say “consumed.” Such as: “…the diet consumed by the goats….”
l. 245: Same issue as 139-142 in terms of low SCC.
Tables 1-3: It may be appropriate to give the definitions of LSM and SE before explanation of the following abbreviations, since those abbreviations are used first.
l. 299: Same consideration as l. 245 for the statement “average SCC was extremely low……”
Table 3: There appears to be an error for TS (%) for the Experimental Group, end value. Also, could the Urea value for Experimental Group, Beginning be an error?
ll. 349-365, whole paragraph: Could some of these differences relate to the use of different analytic methods?
l. 375: For “….in our study indicates…” You might consider being more conservative and use “suggests” or something like that.
l. 413: Do you mean “defensins” here rather than “defensings”?
I did not check the new references in terms of whether they were cited in the text. You should make sure you that.
Author Response
The authors thanks Reviewer for very valuable comments and suggestions. We did our best to correct the manuscript properly.
Please, find attached the file with our responses
